# In Search for an Ionic Liquid with the Best Performance during ^210^Pb/^210^Bi Cherenkov Counting in Waters on an LS Counter

**DOI:** 10.3390/ijerph192416563

**Published:** 2022-12-09

**Authors:** Ivana Stojković, Milan Vraneš, Teona Teodora Borović, Nataša Todorović, Jovana Nikolov, Milka Zečević, Slobodan Gadžurić, Simona Mancini

**Affiliations:** 1Faculty of Technical Sciences, University of Novi Sad, Trg Dositeja Obradovića 6, 21000 Novi Sad, Serbia; 2Department of Chemistry, Biochemistry and Environmental Protection, Faculty of Sciences, University of Novi Sad, Trg Dositeja Obradovića 3, 21000 Novi Sad, Serbia; 3Department of Physics, Faculty of Sciences, University of Novi Sad, Trg Dositeja Obradovića 3, 21000 Novi Sad, Serbia; 4Laboratory “Ambients and Radiations (Amb.Ra.)”, Department of Computer Engineering, Electrical Engineering and Applied Mathematics (DIEM), University of Salerno, 84084 Fisciano, Italy

**Keywords:** ionic liquids, ^210^Pb/^210^Bi detection, Cherenkov counting, Quantulus 1220

## Abstract

The research presented in this paper aims to investigate the performance of several newly synthesized ionic liquids during ^210^Pb/^210^Bi detection in water on a liquid scintillation spectrometer Quantulus 1220 via Cherenkov counting. These experiments have been triggered by the recent reports that certain ionic liquids can act as wavelength shifters, thus significantly increasing the detection efficiency of Cherenkov radiation. The benefit of ionic liquid’s addition to the analysed samples is reflected in the detection limit’s decrement during ^210^Pb quantification, which is pertinent considering naturally low levels of ^210^Pb in aqueous samples. Firstly, it was discovered that ionic liquid, 1-butyl-3-methylimidazolium salicylate, is more efficient than the previously explored 2-hydroxypropylammonium salicylate. Consequently, the impact of a few other ionic liquids on Cherenkov counting efficiency with the same cation group (1-butyl-3-methylimidazolium benzoate, 1-butyl-3-methylimidazolium 3-hydroxybenzoate and 1-butyl-3-methylimidazolium 4-hydroxybenzoate) was also explored to test their potential influence. Molecular simulations have been carried out to reveal which structures of ionic liquids assure wavelength-shifting behavior. The obtained results confirmed that, among the investigated ones, only ionic liquids with the salicylate anion exhibited a wavelength shifting effect.

## 1. Introduction

The occurrence of ^210^Pb ($T_{1/2}$ = 2.26 y) in aquatic environments originates from ^238^U decay series, and its determination continues to evoke interest in various scientific studies. At first, its presence in waters poses a significant radiological risk. Consequently, its determination is included in the international legislations for the radiological assessment of drinking water, recommending its maximum acceptable concentration of 0.1 Bq l^−1^ [1]. Additionally, ^210^Pb and ^210^Po distribution in the marine environment has been extensively used to determine the removal rates of particles from the ocean and particle fluxes during transport along the coast [2,3], as well as the particulate organic carbon (POC) export in the upper ocean [4].

Determination of ^210^Pb content in waters is not a trivial task since its environmental concentrations are very low. At the same time, its decay scheme assumes weak gamma transitions and the emissions of low-energetic beta particles [5]. Its direct detection and quantification are possible by gamma spectrometry. Still, it is time-consuming, and it yields relatively high detection limits, with a large sample volume required for the analysis and the high self-absorption of soft gamma-rays in the sample and the detector [5,6]. 

There are also two possibilities of indirect ^210^Pb measurement by detecting its progenies ^210^Po and ^210^Bi, if they are in radioactive equilibrium with ^210^Pb. Determination of ^210^Po via alpha spectrometry can be performed after two spontaneous depositions of ^210^Po on a silver foil with sufficiently low detection limits, but it demands an in-growth period of several months for ^210^Po/^210^Pb equilibrium to take place, introducing long delays in between sampling separation and counting [7]. ^210^Bi separation from ^210^Pb is usually followed by its activity determination via Liquid Scintillation Counting (LSC). It provides faster analysis (the required storage periods are shorter since ^210^Bi/^210^Pb equilibrium is reached within 40 days), and the acceptable detection limits (several times higher than alpha spectrometry, but still two orders of magnitude lower than the ones for gamma spectrometry). The drawbacks of LSC methods are the need to correct chemical and/or colour quench effects that occur in the samples and cause erroneous results, and, in the case of simultaneous alpha/beta detection, the precise determination of ^210^Pb counting efficiency is challenging because the alpha/beta spectra subsequent separation is not 100% efficient [7].

One other possibility is to measure ^210^Pb on an LS counter if it is in equilibrium with its progeny ^210^Bi through the Cherenkov counting method. Namely, high-energy beta emitter ^210^Bi ($T_{1/2}$ = 5.012(5) d, $E_{\beta max\;}$ = 1162.2(8) keV) emits electrons that have enough energy to produce Cherenkov radiation in water [8]. Cherenkov counting advantage lies in easy sample preparation since ^210^Bi separation from ^210^Pb is not necessary (the electrons emitted from ^210^Pb are not able to generate Cherenkov radiation) but has the drawback of lower counting efficiencies than standard LSC methods, typically from 10% [9] up to 20% [10]. Unlike LSC, the generated Cherenkov spectrum is unaffected by the chemical quenching since no scintillation cocktail is added to the sample [11] and has a lower background count rate which provides acceptable detection limits [10].

Recent reports had demonstrated that the excessive Cherenkov radiation signals from ^210^Bi were detected when ionic liquid (IL) 2-hydroxypropylammonium salicylate, [HPA][Sal], was added to small amounts into the counting vial because this IL acted as the wavelength shifter [12]. Namely, 1.4 g of [HPA][Sal] increased the efficiency for Cherenkov counting during ^210^Pb/^210^Bi detection from ~15% to ~60%. Similar research was reported in [13], confirming that one other IL, 1-butyl-3-methylimidazolium 8-hydroxypyrene-1,3,6-trisulfonate, also manifests wavelength-shifting properties, causing the increase of detection efficiency for ^18^F and ^32^P via Cherenkov counting for 124% and 14%, respectively. Another IL, 3-methylpiridinium salicylate, had been reported to exhibit both scintillating and wavelength shifting effects, with a significant influence on ^210^Pb Cherenkov spectra, ^210^Pb and ^226^Ra gross alpha/beta spectra, and even ^3^H spectra generation [14].

The use of ionic liquids as scintillators is desirable due to their unique properties. Ionic liquids are considered as non-volatile solvents [15], which have good thermal and chemical stability [16]. Ionic liquids can be easily recycled and reused in an analytical process without losing efficiency. Combining many cations and anions makes it possible to tune their physical and chemical properties and obtain ionic liquids as an optimal reagent for an industrial or laboratory process [17].

This paper explores several newly synthesized ILs’ impacts on ^210^Pb/^210^Bi Cherenkov counting, offering explanations based on their chemical structure. To better understand the mechanism of wavelength-shifting, the influence of the change of cation and anion structure for five different ionic liquids was investigated. For cations were used 2-hydroxypropylammonium and 1-butyl-3-methylimidazolium, and for an anion salicylate, benzoate, 3-hydroxybenzoate and 4-hydroxybenzoate were used. A well-studied [Bmim]^+^ ion was chosen as the cation to compare whether the presence of an aromatic cation and additional π-π interactions between the cation and hydroxybenzoate affect the scintillating effect. The results were compared with systems containing an aliphatic [HPA]^+^ cation.

Regarding the change in the anionic structure, a fine structural variation of the position of the hydroxyl group of the corresponding hydroxy benzoates was investigated. The variation of the positions of *-ortho*, *-meta* and *-para* in the series of hydroxy benzoates in previous works has shown a significant effect on different physicochemical properties such as the micellization of surface-active substances as well as on the toxicity of ionic liquids [18,19]. The research conducted in this paper provides deeper insights and answers to the question of which IL structures assure wavelength-shifting behaviour.

## 2. Materials and Methods

### 2.1. Instrumentation and Materials

The detection system used for all measurements was Liquid Scintillation Spectrometer Quantulus 1220^TM^, convenient for the precise low-level radioactivity measurements that are often necessary for environmental radioanalysis. Quantulus 1220 is equipped with the passive shield made from lead, asymmetrically distributed around the detector assembly that absorbs cosmic radiation. The addition of the copper head of the piston absorbs X-rays generated in the lead by cosmic radiation interactions. In addition to this, the detector possess an active shield that further reduces cosmic and environmental gamma radiation. This asymmetric guard counter contains the liquid scintillation filling surrounded with two extra photomultipliers operating in the summed coincidence. This active guard shield operates in anticoincidence with the sample detector and its two photomultipliers that surround the measurement chamber with the vial [20]. Cherenkov spectra were acquired and analysed by WinQ and EASYView software.

To provide the lowest background counting rate and the highest detection efficiency, low diffusion polyethylene vials (Super PE vial Cat.No. 6008117) were selected since ^40^K that is present in glass vials induces higher background, whilst the vial’s maximum capacity of 20 mL was kept. The comparison of performance of plastic, glass and low ^40^K glass vials during Cherenkov counting for ^90^Sr detection was done in our previous work [21]. The experiments were carried out with the calibration samples prepared with the standard radioactive source produced by the Czech Metrology Institute, Inspectorate for Ionizing Radiation, an aqueous ^210^Pb solution with a certified activity A(^210^Pb) = 29.55 Bq ml^−1^ and combined standard uncertainty 1.0% on a reference date 1 October 2013.

After the counting, $R_{C}$ [s^−1^] and $R_{0}$ [s^−1^] were obtained as the count rates of calibration (reference standard) and background samples, respectively. Since the reference activity of the calibration sample was A [Bq], the detection efficiency was evaluated from the expression:(1)$$\varepsilon =\frac{R_{C}-R_{0}}{A}\;.$$

### 2.2. IL Synthesis

Chemical structures of all investigated ILs are presented in Table 1. The provenance and purity of used compounds are given in Table A1 in Appendix A and were used as received without further purification. NMR spectra of all synthesized ILs are provided in Figure A1, Figure A2, Figure A3, Figure A4 and Figure A5 in Appendix A.

#### 2.2.1. Synthesis of 1-Butyl-3-Methylimidazolium-Based Ionic Liquids

The preparation of 1-butyl-3-methylimidazolium benzoate, [Bmim][Ben], was accomplished by mixing the equimolar amounts of 1-butyl-3-methylimidazolium chloride and sodium benzoate. Both compounds were dissolved in acetone and stirred in a round-bottom flask for 5 h. The resulting white precipitate (sodium chloride) was removed by filtering under vacuum, and the clear, pale yellow liquid was obtained. The remaining solvent was removed using a rotary evaporator. After achieving a constant mass of [Bmim][Ben], the obtained product was additionally dried under the vacuum. A similar procedure was used for all the other ionic liquids synthesized in this work. Only different starting compounds were used to obtain different ionic liquids. Therefore, instead of sodium benzoate, sodium salicylate, sodium 3-hydroxybenzoate and sodium 4-hydroxybenzoate were used to obtain [Bmim][Sal], [Bmim][3HB] and [Bmim][4HB], respectively. The residual chloride in the samples was tested using a spot-test by AgNO_3_.

#### 2.2.2. Synthesis of Ionic Liquid 2-Hydroxypropylammonium Salicylate

Ionic liquid 2-hydroxypropylammonium salicylate, [HPA][Sal], was prepared in the described way. An equimolar amount of salicylic acid was dissolved in methanol, added dropwise to a 1-amino-2-propanol water solution and cooled in an ice bath. After adding the salicylic acid, the reaction mixture was stirred at room temperature for 2 h. The obtained ionic liquid was dried under vacuum for the next 3 h to remove any traces of methanol and water. The obtained liquid product was stored in a vacuum desiccator over P_2_O_5_ for the next 48 h.

### 2.3. Molecular Simulations

Theoretical predicting and describing of the scintillating effect of the examined ionic liquids by the DFT calculations were applied using Jaguar 9.0 software as a part of Schrödinger Materials Science Suite 2015-4. The B3LYP exchange-correlation functional with empirical correction for dispersion (B3LYP–D3) was used with a 6–31 + G(d,p) basis set. Simulations were performed using Generalized Valence Bond Perfect-Pairing (GVB-PP). This pseudospectral method, extended to electron correlation methods, can predict accurate excitation energies, rotational barriers and bond energies. The Continuum solvation model was used, in particular the Generalized Born model. To ensure the validity of the obtained structures, geometrical optimizations were followed by harmonic frequency analysis, where structures without imaginary wavenumbers were considered for further investigation. The intermolecular non–covalent interactions (NCI) have been examined using the method described in the literature [22]. 

HOMO and LUMO orbital energies were calculated using monitored surfaces (molecular orbitals, density, potential) and the potential charge of the atomic electrostatics. HOMO energy suggests the region of the molecules which can donate an electron, while LUMO energy signifies the capacity of the molecule to accept the electrons. The difference in HOMO and LUMO energy, also known as HOMO–LUMO gap energy, indicates the electronic excitation energy necessary to compute the stability of the compounds and can describe the scintillating potential of a substance. 

The Fukui functions are partial derivatives of the electron and spin density concerning a change in either the electron count or the unpaired spin count. A change in the electron count can result from a reaction with another molecule or by any external charge transfer mechanism. A change in the number of the unpaired electron spins would be induced by electromagnetic radiation to produce an electronically excited state of different spin multiplicity. Hence, the scintillating potential of a compound can be predicted from Fukui functions that describe spin density:(2)$$f^{+}=\frac{\rho^{N+\delta }\left(r\right)-\rho^{N}\left(r\right)}{\delta },$$
(3)$$f^{-}=\frac{\rho^{N-\delta }\left(r\right)-\rho^{N}\left(r\right)}{\delta },$$
where N is the number of electrons in the reference state of the molecule, and δ is a fraction of an electron.

## 3. Results and Discussion

### 3.1. IL’s Influence on the Detection Efficiency of Cherenkov Counting

ILs presented in Table 1 were added in the increasing mass to the prepared sets of ^210^Pb calibration samples. All samples contained the same ^210^Pb activity concentration (A = 4.95(7) Bq) spiked to distilled water so that the total sample volume was kept at 20 mL. Before ILs’ addition, all samples were stored for 50 days to assure that the secular equilibrium between ^210^Pb and ^210^Bi was reached, after which they were counted in several cycles for 100 min. Background samples were used to determine MDA (Minimal Detectable Activity) that decreased with the measurement time and were prepared with 20 mL of distilled water transferred into the vial. 

Cherenkov radiation detection was carried out through the counting protocol set up manually on Quantulus 1220, the setup configuration was given in the previous research [21]. The only difference in the counting protocol was coincidence bias selection which was set to low instead of high since it was determined that the electrons emitted from ^210^Bi generated Cherenkov spectrum with better resolution and higher sensitivity on low coincidence bias [12]. According to the expression (1), the obtained count rates were used for the detection efficiency evaluation.

It was reported earlier that [HPA][Sal] efficiently increases the detection efficiency of Cherenkov counting [12]. This is the reason that the first IL that was synthesized was [Bmim][Sal], which also contained salicylate as an anion like [HPA][Sal]. The obtained detection efficiency is presented in Figure 1a, together with the previously published results for [HPA][Sal] addition. In this way, it was possible to compare these two ILs’ effects on Cherenkov spectra and derive some assumptions on which chemical structure acts more efficiently. It is clear that when the amount of the added IL exceeds 0.5 g, [Bmim][Sal] becomes more efficient than [HPA][Sal], causing a more significant efficiency increment, Figure 1a.

Applying [Bmim][Sal] during ^210^Pb detection in the water samples via the Cherenkov counting method is a significant discovery since natural ^210^Pb activity concentrations are extremely low. Therefore, it necessitates precise and sensitive methods for its quantification. According to Currie relation [23], if the detection efficiency is higher, it linearly decreases MDA. Therefore, the increment in the detection efficiency from 16% to >70% in the presence of small amounts (around 0.9 g) of [Bmim][Sal] can reduce the detection threshold by more than four times and offers innovative and unmatched improvement in the existing methods. 

The explanation in the increment of the count rates with the addition of both ILs will be elaborated in the following paragraphs.

The threshold for Cherenkov photon production varies with the refractive index of a transverse medium. If this parameter is higher, the required energy for Cherenkov radiation generation is lowered, which would cause an excess in the obtained count rates [24]. However, this explanation could not be maintained since the negligible amounts (approximately up to 1 g) of ILs added to 20 mL of water could not significantly alter the sample’s refractive index.

On the other hand, it was noticed that the addition of ILs did not alter the shape or position of Cherenkov spectra, suggesting that ILs could act as wavelength shifters. A substantial fraction of the emitted Cherenkov photons have energies in the ultraviolet region, which are not detectable by the photomultiplier tubes that operate inside an LS counter [25]. ILs absorb ultraviolet photons and re-emit them at longer wavelengths, thus inducing the wavelength shifting effect. Consequently, the photomultiplier tubes detect more light and the counting efficiency increases.

Another explanation for the efficiency increment could be that ILs induce a scintillating effect. This was investigated when spectra of the samples containing only ^210^Pb standard and the samples of ^210^Pb standard + maximal amount of the added ILs were inspected after their counting on the default gross alpha/beta counting protocol on Quantulus 1220. Namely, the gross alpha spectrum was not generated in any case. If ILs acted as a scintillator, ^210^Po alpha spectrum (^210^Pb’s progeny) should appear to some extent in the samples that contain ILs. 

The conclusion on the two investigated ILs’ behaviour is that their presence does not influence the sample’s refractive index and cannot affect the Cherenkov threshold. These two ILs do not exhibit scintillating properties (if they acted like scintillators, there is no explanation why ^210^Po alpha spectrum was not generated in their presence). Cherenkov emissions from ^210^Bi generated excessive signals in the spectra with ILs’ addition due to the spectral wavelength shifting effect. 

The fact that [Bmim][Sal] had better performance than [HPA][Sal] during Cherenkov detection induced a hypothesis that 1-butyl-3-methylimidazolium cation structure is responsible for its greater efficiency. This possible explanation had to be tested, thus triggering another research, a synthesis of a few novels ILs with the same cation, 1-butyl-3-methylimidazolium-benzoate, [Bmim][Ben], 1-butyl-3-methylimidazolium 3-hydroxybenzoate [Bmim][3HB] and 1-butyl-3-methylimidazolium 4-hydroxybenzoate [Bmim][4HB]. The sets of samples were prepared in the same manner as the previous calibration samples, and were spiked with the increasing amounts of these ILs. The obtained efficiencies for all four ILs with the same cation structure are displayed in Figure 1b. 

Since no other IL except [Bmim][Sal] caused the detection efficiency increment, it should be stated that the cation structure itself does not influence the count rate increment. It is interesting to note that [Bmim][3HB] and [Bmim][4HB] even caused a mild decrement in detection efficiency. They induce colour quench, to which Cherenkov counting is sensitive and demands corrections. Namely, Cherenkov pulse height distribution is being shifted towards lower energy channels in the spectrum. It is generated with lesser intensity in the coloured samples, decreasing the obtained count rate and the counting efficiency [26].

#### Reproducibility Investigation

Finally, reproducibility was considered a dispersion of the detection efficiencies obtained in replicates of calibration samples prepared with the same ^210^Pb activity concentration spiked to counting vials, but with [Bmim][Sal] that was synthesized twice. The set with IL that was firstly synthesized was denoted as the first experiment, while the second experiment assumed the addition of IL that was synthesized after the first one. Approximately the same amount of the increasing mass of [Bmim][Sal] was added to two sets of calibration samples. The results of measurements are displayed in Figure 2.

The graph in Figure 2a shows the ratio $\varepsilon /\varepsilon_{0}$ for the different mass addition of [Bmim][Sal] to two sets of samples (the added amounts could not be precisely equal between the sets), where ε is Cherenkov detection efficiency obtained in the presence of a certain mass of [Bmim][Sal] and $\varepsilon_{0}$ is Cherenkov detection efficiency in spiked ^210^Pb samples without the addition of [Bmim][Sal], $\varepsilon_{0}$ ≈ 16%. All results are mutually consistent within the measurement uncertainty for the two samples with similar masses of added [Bmim][Sal]. Furthermore, the correlation function between the obtained detection efficiencies in Figure 2b is rather satisfying, $y=1.11\left(6\right)x-0.029\left(19\right)$, with the correlation coefficient close to 1. This value, 1.11(6) was obtained for the samples that do not contain precisely the same mass of [Bmim][Sal], and it would be even closer to 1 if the added mass matched better between the sets of samples. Therefore, it can be concluded that the method with the addition of ILs during Cherenkov counting provides results with excellent reproducibility.

### 3.2. Results of Molecular Simulations

The DFT calculations were performed to examine the scintillating activity of synthesized ionic liquids. To obtain a more realistic simulation of the transition state geometry, all simulations were performed using the generalized valence bond settings. Figure 3 shows optimized geometrical structures with the representation of noncovalent interactions. 

The charge density around all ionic liquids used in this work is presented in Figure 4. From Figure 4, it can be seen that the blue regions represent positive charge, along with the red areas representing negative charge.

Substances with good scintillating properties are substances that can efficiently produce an excited state. This is usually characteristic of molecules with accessible excited states at lower energies and non-bonded π electrons that can easily be excited. Computational descriptors are good for explaining the excitation potential of molecules by predicting energy for the highest occupied molecular orbital (HOMO) and the lowest unoccupied molecular orbital (LUMO). The values of HOMO and LUMO energies together with energy gaps ($\Delta E_{gap}$) are presented in Table 2. The energy gaps are calculated with the following equation:(4)$$\Delta E_{gap}=E_{\mathrm{LUMO}}-E_{\mathrm{HOMO}}.$$

The rigidity of molecules is significant for wavelength shifters. It is achieved either by condensing aromatic molecules or connecting them to allow efficient delocalization of electrons via unhybridized 2pz atomic orbitals [27]. In the ILs’ case, the structural rigidity of the ion pair can be achieved by increasing the number of non-covalent interactions between cation and anion. The rigidity of the molecule and the efficiency of electron delocalization through the system directly affect the values of the HOMO and LUMO orbitals energies. Our previous research [14] has shown that the energy gap between the HOMO and LUMO orbitals (energy gap) significantly impacts the scintillating and wave-shifting ability of compounds and can be an important parameter for evaluating the efficiency of a compound in an LS counter.

As shown in Table 2, the HOMO-LUMO energy gaps of [Bmim][Sal] ionic liquids are slightly higher than those of other ionic liquids. This data indicates that [Bmim][Sal] demands more energy to become excited and emit a photon. The HOMO and LUMO orbitals were performed and shown in Figure 5 and Figure 6 to understand better and further illustrate differences in the electron structure of all ionic liquids investigated in this research.

The Atomic Fukui indices were calculated to distinguish and quantify differences in HOMO and LUMO orbitals more precisely. From Fukui indices, it can be seen that each index includes two subscripts that can take the values N or S, which represent the electron density and the spin density, respectively. The nature of the partial derivative on which the index is based is indicated with two indices. The first index indicates the property that responds to a change in the property indicated by the second index. According to this, f_NS indicates the change in electron density about an atom when the molecule undergoes a reaction in which its spin multiplicity changes. It is established as a descriptor of how easily a molecule can be excited. If the value of an index is higher, the higher is the change in electron or spin density near the atom of interest. The enumeration of ionic liquids and analysis of HOMO f_NS and LUMO f_NS indices is presented in Figure 7 and Figure 8.

From Figure 8, it can be seen that [HPA][Sal] ionic liquid has higher and positive values of HOMO orbitals than those of the other ionic liquids. The most positive f_NS values indicate the largest changes in the electron density at HOMO orbitals of this ionic liquid when a reaction occurs where the shape of the spin changes. It is more important to pay attention to the values of LUMO f_NS orbitals. By comparing ionic liquids with the same cation and a different anion, it is concluded that [Bmim]^+^ has better properties as a cation because the whole ring has positive LUMO f_NS values. This means that the ring of [Bmim]^+^ cation can be excited, which is not the case for [HPA]^+^ cation. If we compare ionic liquids with the same cation and different anion, it can be seen that [Ben]^−^ anion, which does not have a hydroxyl group, is inert in terms of the electron density. 

Additionally, from Figure 8, it can be seen that most values of LUMO f_NS for [Bmim][Ben] are near zero. Additionally, from Figure 8, we can see that the values of LUMO f_NS are concentrated around the oxygen atom of the carboxyl group (numeration O19) with a pronounced negative value which indicates that in the case of this ionic liquid excitation will not be good. Significant changes occur with introducing the hydroxyl group into the anion structure. In the case of ionic liquid with [Sal]^−^ anion, LUMO f_NS values for the oxygen atom (O19) are slightly positive. From this obtained data, it can be concluded that [Bmim][Sal] ionic liquid is the most active scintillator because all LUMO f_NS values around the cation ring are positive. Results obtained from molecular simulations are in accordance with the previous experimental results.

## 4. Conclusions

This work explored the performance of several newly synthesized ionic liquids during ^210^Pb/^210^Bi Cherenkov counting on an LS counter Quantulus 1220, explaining their behaviour based on ILs’ structure. Among the few ILs with the same cation structure, 1-butyl-3-methylimidazolium [Bmim]^+^, the only one that significantly increased the detection efficiency was [Bmim][Sal], which contained salicylate as an anion. Moreover, it was shown that [Bmim][Sal] had a more significant impact on the efficiency in comparison with the previously investigated 2-hydroxypropylammonium salicylate, [HPA][Sal]. Other ILs, the one with benzoate anion, [Bmim][Ben], did not increase the obtained count rates, while the structures with 3-hydroxybenzoate and 4-hydroxybenzoate anions, [Bmim][3HB] and [Bmim][4HB], respectively, induced mild colour quench, even reducing the initial detection efficiency. ILs’ behaviour could be explained via analysis of their HOMO f_NS and LUMO f_NS values. The presented research confirmed that salicylates act as wavelength shifters, consequently increasing the detection efficiency of Cherenkov counting. This finding can be very useful during the detection of naturally low levels of ^210^Pb in waters via the Cherenkov counting method, since the increment in detection efficiency in the presence of small amounts of [Bmim][Sal], about 0.9 g, can reduce detection threshold more than four times. Moreover, ILs could be applied in quantifying other radionuclides besides ^210^Pb/^210^Bi via Cherenkov counting. Further research on the synthesis of ionic liquids with a more extensive range of HOMO-LUMO orbitals, i.e., energy gaps, is needed to identify the potential limit value required for the wavelength shifting effect.

## Figures and Tables

**Figure 1 ijerph-19-16563-f001:**
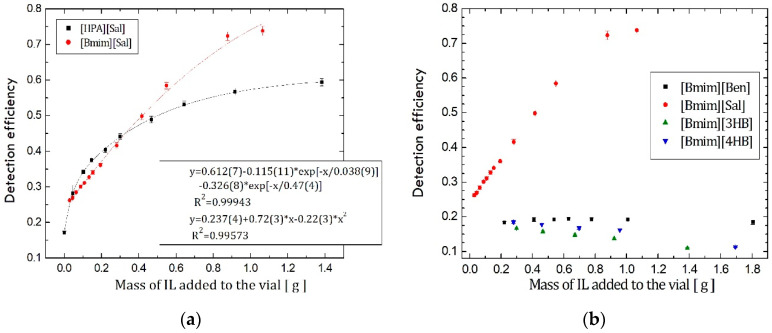
Comparison of IL’s influence on ^210^Pb/^210^Bi detection efficiency: (**a**) [HPA][Sal] and [Bmim][Sal]; (**b**) Several ILs with [Bmim]^+^ cations.

**Figure 2 ijerph-19-16563-f002:**
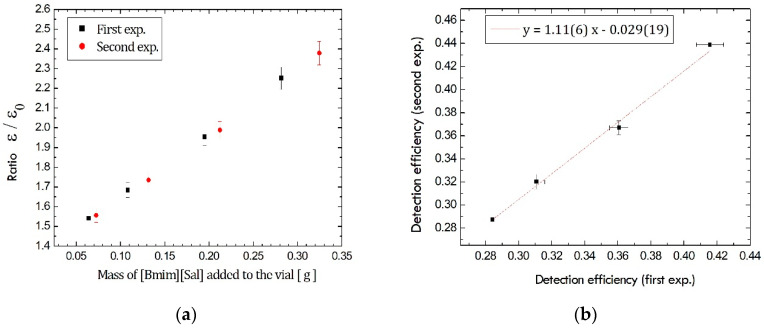
Reproducibility tests for the two independent experiments: (**a**) Efficiency ratio for similar addition of [Bmim][Sal]; (**b**) Correlation between the obtained efficiencies.

**Figure 3 ijerph-19-16563-f003:**
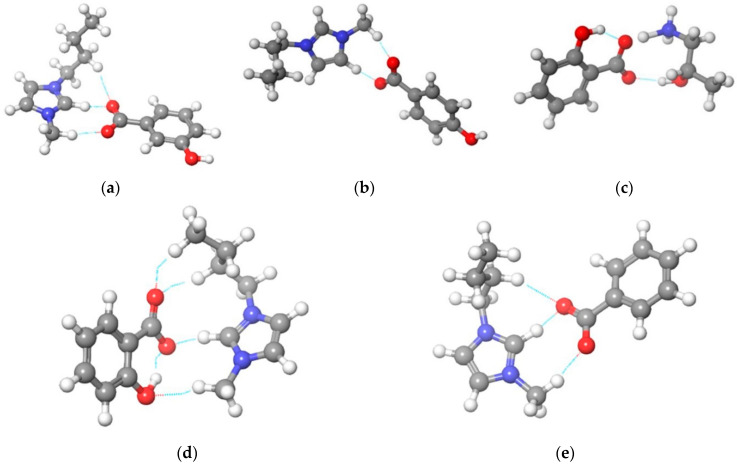
Optimized structures with a representation of noncovalent interactions for: (**a**) [Bmim][3HB]; (**b**) [Bmim][4HB]; (**c**) [HPA][Sal]; (**d**) [Bmim][Sal]; (**e**) [Bmim][Ben].

**Figure 4 ijerph-19-16563-f004:**
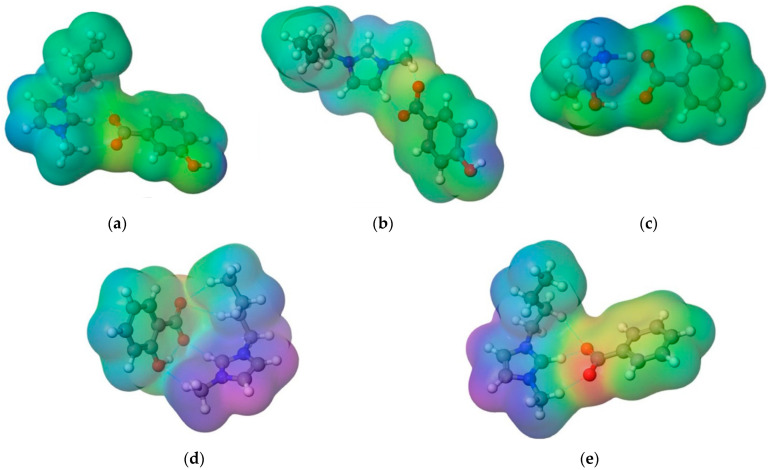
The pictorial representation of a charge density for: (**a**) [Bmim][3HB]; (**b**) [Bmim][4HB]; (**c**) [HPA][Sal]; (**d**) [Bmim][Sal]; (**e**) [Bmim][Ben].

**Figure 5 ijerph-19-16563-f005:**
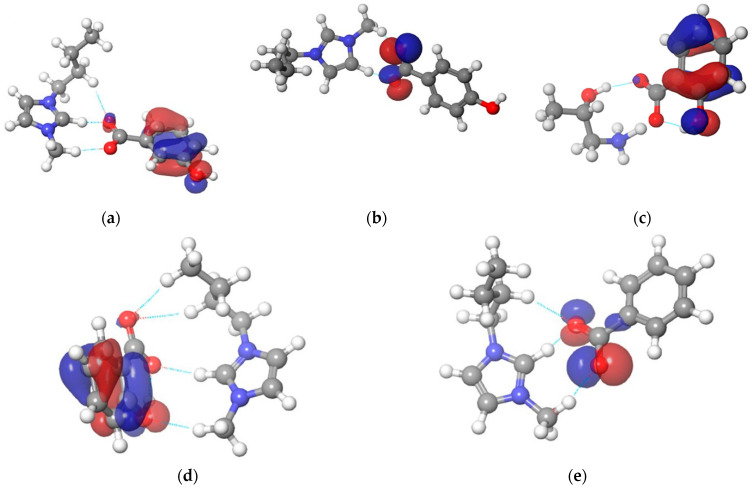
Representation of HOMO orbitals of: (**a**) [Bmim][3HB]; (**b**) [Bmim][4HB]; (**c**) [HPA][Sal]; (**d**) [Bmim][Sal]; (**e**) [Bmim][Ben].

**Figure 6 ijerph-19-16563-f006:**
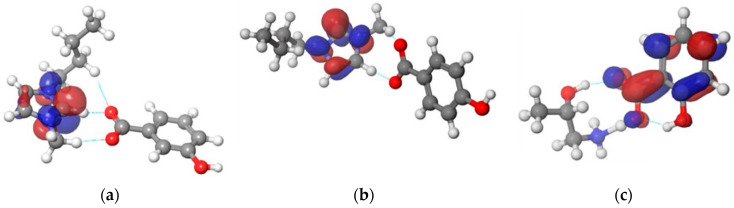
Representation of LUMO orbitals of: (**a**) [Bmim][3HB]; (**b**) [Bmim][4HB]; (**c**) [HPA][Sal]; (**d**) [Bmim][Sal]; (**e**) [Bmim][Ben].

**Figure 7 ijerph-19-16563-f007:**
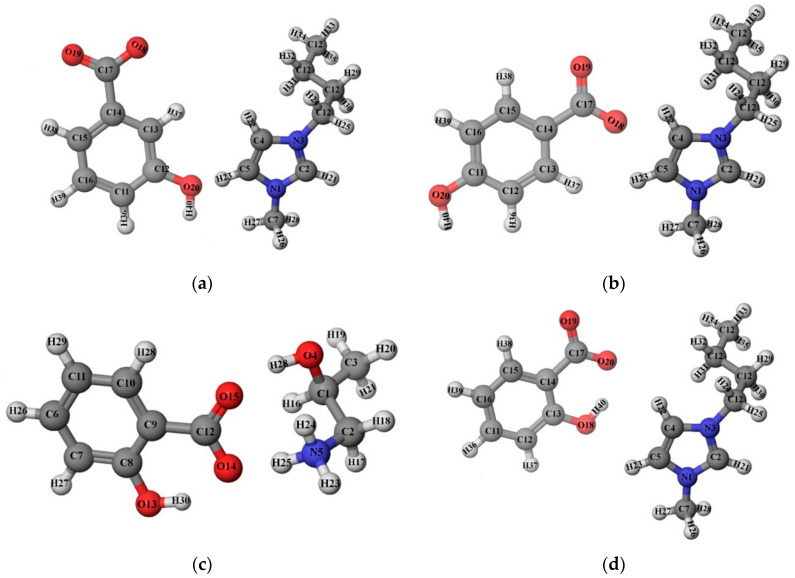
The structure of ionic liquids and the atom-numbering scheme of: (**a**) [Bmim][3HB]; (**b**) [Bmim][4HB]; (**c**) [HPA][Sal]; (**d**) [Bmim][Sal]; (**e**) [Bmim][Ben].

**Figure 8 ijerph-19-16563-f008:**
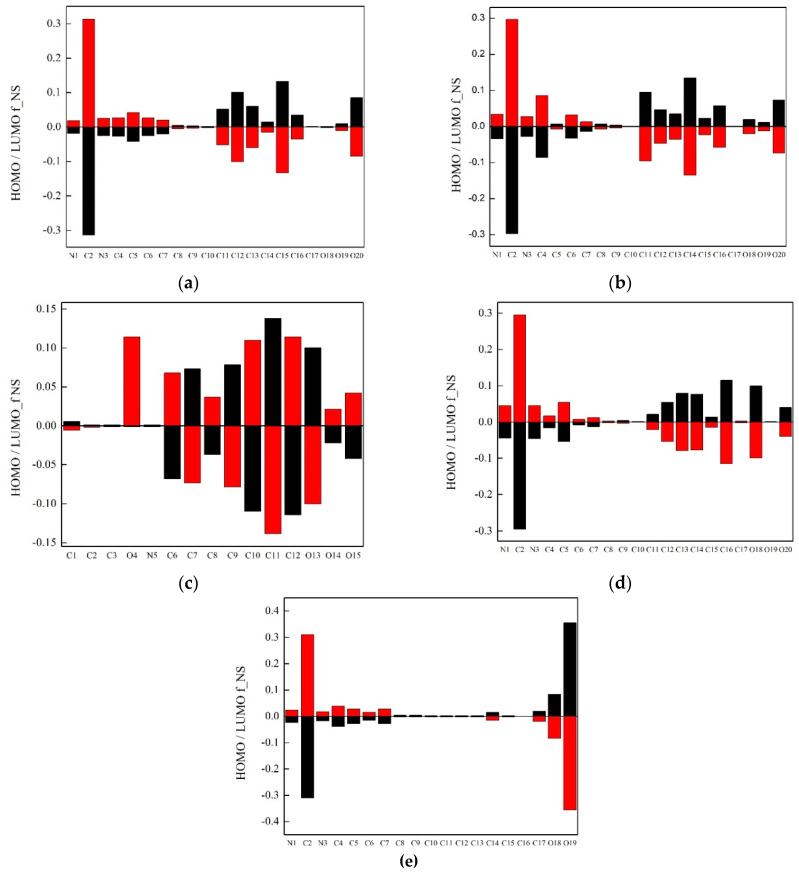
Representation of Fukui indices (black represents HOMO f_NS and red represents LUMO f_NS): (**a**) [Bmim][3HB]; (**b**) [Bmim][4HB]; (**c**) [HPA][Sal]; (**d**) [Bmim][Sal]; (**e**) [Bmim][Ben].

**Table 1 ijerph-19-16563-t001:** Chemical structures of synthesized ionic liquids.

Ionic Liquid	Chemical Structure
1-butyl-3-methylimidazolium 3-hydroxybenzoate [Bmim][3HB]	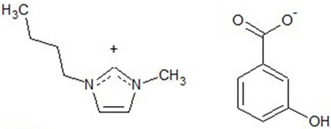
1-butyl-3-methylimidazolium 4-hydroxybenzoate [Bmim][4HB]	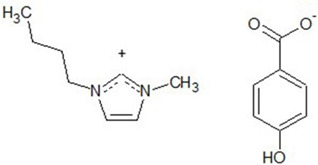
2-hydroxypropylammonium salicylate [HPA][Sal]	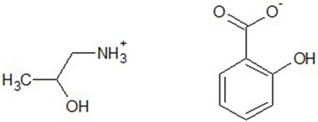
1-butyl-3-methylimidazolium salicylate [Bmim][Sal]	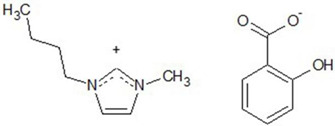
1-butyl-3-methylimidazolium benzoate [Bmim][Ben]	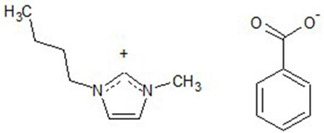

**Table 2 ijerph-19-16563-t002:** HOMO/LUMO energies and energy gaps for [Bmim][3HB], [Bmim][4HB], [HPA][Sal], [Bmim][Sal], and [Bmim][Ben].

Compound	$E_{HOMO}\;[eV]$	$E_{LUMO}\;[eV]$	$\Delta E_{gap}\;[eV]$
[Bmim][3HB]	−5.123	−0.520	4.603
[Bmim][4HB]	−4.791	−1.179	3.612
[HPA][Sal]	−6.002	−1.075	4.927
[Bmim][Sal]	−6.024	−0.778	5.245
[Bmim][Ben]	−5.292	−0.537	4.755

## Data Availability

The data presented in this study are available on request from the corresponding author.

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
