# Peer review of "In Search for an Ionic Liquid with the Best Performance during 210Pb/210Bi Cherenkov Counting in Waters on an LS Counter"

_ijerph, 2022, doi:10.3390/ijerph192416563_

Round 1

Reviewer 1 Report

Dear Author/s,

The manuscript is well-written and could address all matters. not much correction required just small comment.

find attached file my comments.

Thank you.

Author Response

Please, see the attached file

Reviewer 2 Report

1) line  72, Cerenkov should be changed to Cherenkov

2) lines 247, 248. It is unclear how a beta spectrum can be formed if the scintillator is without a scintillator

3) The calculations are not sufficiently discussed for specialists in the field of chemistry and activity detection. A real comparison of these HOMOs and LUMOs and the difference between them with the IL structure and the demonstrated detection efficiency did not presented. 

Author Response

Please, see the attached file
